# Neurotensin and Its Involvement in Reproductive Functions: An Exhaustive Review of the Literature

**DOI:** 10.3390/ijms24054594

**Published:** 2023-02-27

**Authors:** Pauline Arbogast, Guillaume Gauchotte, Romane Mougel, Olivier Morel, Ahmed Ziyyat, Mikaël Agopiantz

**Affiliations:** 1Department of Reproductive Medicine, CHRU de Nancy, Université de Lorraine, F-54000 Nancy, France; 2Department of Pathology, CHRU de Nancy, Université de Lorraine, F-54500 Vandœuvre-lès-Nancy, France; 3INSERM UMRS 1256, Nutrition-Genetics and Exposure to Environmental Risks (NGERE), Université de Lorraine, F-54000 Nancy, France; 4Department of Gynecology and Obstetrics, CHRU de Nancy, Université de Lorraine, F-54000 Nancy, France; 5INSERM U1016, CNRS UMR8104, Institut Cochin, Université de Paris, F-75014 Paris, France; 6Department of Histology, Embryology and Reproductive Biology, AP-HP, Hôpital Cochin, F-75014 Paris, France

**Keywords:** neurotensin, reproduction, ovulation, acrosome reaction, fertilization, spermatozoa

## Abstract

Neurotensin (NTS) is a peptide discovered in 1973, which has been studied in many fields and mainly in oncology for its action in tumor growth and proliferation. In this review of the literature, we wanted to focus on its involvement in reproductive functions. NTS participates in an autocrine manner in the mechanisms of ovulation via NTS receptor 3 (NTSR3), present in granulosa cells. Spermatozoa express only its receptors, whereas in the female reproductive system (endometrial and tube epithelia and granulosa cells), we find both NTS secretion and the expression of its receptors. It consistently enhances the acrosome reaction of spermatozoa in mammals in a paracrine manner via its interaction with NTSR1 and NTSR2. Furthermore, previous results on embryonic quality and development are discordant. NTS appears to be involved in the key stages of fertilization and could improve the results of in vitro fertilization, especially through its effect on the acrosomal reaction.

## 1. Introduction

Neurotensin (NTS) is a 13 amino acid peptide initially described in 1973 in the bovine hypothalamus [1]. It is synthesized from a 170 amino acid longform propeptide. The human gene for neurotensin is located on chromosome 12q2.1 [2], and its expression is mainly described in the neurons and N endocrine cells in the small intestine [3].

NTS has a physiological role and is implicated in several functions, mainly in digestive and central nervous systems. In the digestive system, NTS acts on intestinal motility [4] and facilitates fatty acid translocation [5]. In the central nervous system, NTS modulates dopaminergic, serotoninergic, GABAergic, glutamatergic, and cholinergic systems [6]. It plays a major role in decreasing body temperature and sleeping [7]. NTS also presents antinociceptive effects mediated by NTS receptor 2 (NTSR2) [8,9].

NTS effects are mediated by three receptors (NTSR): NTSR1 and NTSR2 are G-protein-coupled receptors with seven transmembrane-spanning domains, and NTSR3 or sortilin (SORT1) is a Vps10p receptor family member, which is characterized by the presence of an extracellular region containing a cysteine-rich domain and only one transmembrane domain [10]. NTSR1 and NTSR3 have a strong affinity with NTS [11]. NTSR1 activation stimulates intracellular second messengers (inositol triphosphate (IP3) and diacylglycerol (DAG)). IP3 increases the concentration of intracellular calcium, and DAG induces the stimulation of protein kinase C (PKC) [12,13], which activates mitogen-activated protein kinases (MAPKs)/Erk pathways [14,15].

The NTS/NTSR1 complex is involved in the progression of many cancers, such as colic adenocarcinoma [16], small-cell lung cancer [17], medullary thyroid cancer [18], hepatocellular carcinoma [19], pancreatic carcinoma [20], breast cancer [21], and non-small-cell lung cancer [22], via cell proliferation, survival, migration, invasion, and neoangiogenesis [23]. Our team has shown that NTSR1 expression levels are correlated with the grade and prognosis of cancer in several tumors, especially in endometrial and ovarian adenocarcinomas, with its cytoplasmic localization increasing with the grade of the tumor [24,25].

Several publications have reported the expressions of NTS and its receptors in reproductive tissues, related to their roles in reproduction functions, particularly in ovulation, sperm capacitation, fertilization, and embryonic development. It has recently been highlighted that NTS receptors are located on male gametes and that NTS is expressed in the uterus. NTS could play a role in reproduction issues by participating in follicular rupture during ovulation [26,27] and improving sperm capacitation and the acrosome reaction [28,29]. Moreover, after fertilization, NTS would improve the rate of embryonic cleavage and embryo quality in cattle and mice [30,31]. 

The aim of this review was to synthetize the worldwide knowledge about NTS and its potential role in reproductive functions in order to achieve the following:Synthesize knowledge regarding the expression and function of the neurotensinergic complex in the different models in which it has been studied;Propose an integrative mechanistic interpretation;Open perspectives for future research.

On the PubMed database, the keywords “fertility”, “fertilization”, “ovulation”, “oocyte”, “embryo”, “endometrium”, “spermatozoa”, “oviduct”, and “reproduction” associated with NTS were tested. After abstract studies, we considered 15 publications of interest [24,26,27,28,29,30,31,32,33,34,35,36,37,38,39,40,41,42].

## 2. Neurotensinergic Complex Expression in Reproductive System

### 2.1. Oviduct Epithelium and Endometrium

NTS was described in the female tract in the oviduct epithelium using Western Blot, immunohistochemistry, and RT-PCR [28,29,33,39]. Moreover, NTS mRNA levels in female mice treated with equine chorionic gonadotropin (eCG) and human chorionic gonadotropin (hCG) were significantly increased in the ampulla compared to the endometrium and the tubal isthmus epithelial cells [29]. Using a microarray analysis, the expression of NTS mRNA is more than 20 times higher in the isthmus oviduct and 32 times higher in the ampulla during the follicular phase of the cycle than during the luteal phase [43]. NTS expression has also been described in the endometrium in rats [37], mice [29], and cattle [32,38]. Sakumoto et al. studied the expressions of several proteins using immunohistochemistry, including NTS, in the endometrium of cows in summer and fall [32]. The NTS mRNA expression in the endometrium of the cows was more abundant in summer than in autumn. In humans, our team demonstrated using immunohistochemistry and RT-PCR that NTS expression was very weak in the normal endometrium, regardless of the menstrual cycle. NTSR1 was not expressed in the human normal endometrium [24].

### 2.2. Myometrium

Some authors have been interested in the presence of NTS/NTSR1 in the tissues of the human myometrium and leiomyoma and, more particularly, in women having hormonal treatment for in vitro fertilization (IVF) [34]. Rodriguez et al. used in situ hybridization (ISH) to detect NTS mRNA. NTS was present in the connective tissue cells of the normal myometrium. Considering leiomyoma, NTS immunoreactivity has also been detected in smooth muscle cells (up to 14.4%), and its rate is interestingly increased (33.8%) when women have been previously treated for IVF. To test the expression of NTSR1, this team used RT-PCR and detected its expression in connective tissue cells. No significant difference between normal and tumor tissue was found, but NTSR1 levels were also increased in leiomyoma smooth muscle cells after controlled ovarian hyperstimulation (COH). They also showed, using immunohistochemistry, that normal myometrium connective tissue cells only express NTS or NTSR1 but not both, unlike tumor tissue, in which they are both co-expressed [34,44]. 

### 2.3. Cumulus and Granulosa Cells

Hiradate et al. showed using RT-PCR that NTS mRNA is also expressed in the ovulated cumulus cells of female mice. Its expression was massively increased after stimulation by gonadotropin and hCG, as observed in rats and macaques [26,27]. After in vitro maturation (IVM) with FSH and epidermal growth factor (EGF), the expression of NTS mRNA was higher in mature cumulus cells than in immature cumulus cells [29]. The addition of a MAPK inhibitor (U0126) to the medium completely blocked the release of NTS from cumulus cells, which could suggest that the secretion of NTS is regulated by these two factors via the activation of the MAPK pathway [29,45]. In cattle, it has been shown, using RT-PCR, that *cumulus oophorus* and *corona radiata* granulosa cells do not express NTSR1 and NTSR2; NTSR3 was not studied [30]. However, in women and female rats, Al Alem et al. found expressions of the three specific NTS receptors using RT-PCR and immunohistochemistry in granulosa cells, with a major predominance of NTSR3 compared to NTSR1 and 2, which were very weakly expressed [27]. The same result was found in female macaques using RT-PCR, immunohistochemistry, and Western Blot [26]. NTSR3 immunodetection was low to nondetectable in the granulosa cells of ovulatory follicles before hCG and 12 h after hCG administration. Granulosa cell staining for NTSR3 was present 24 h after hCG, with an apparent decline in NTSR3 immunodetection observed by 36 h after hCG. Granulosa cell NTSR3 mRNA showed a similar pattern, with low mRNA levels 0–12 h after hCG, peak NTSR3 mRNA levels 24 h after hCG, and a return to low NTSR3 mRNA levels 36 h after hCG [26]. In female rats, hCG injection did not change the expression levels of NTSR3 in granulosa cells [27]. In mice, NTSR3 expression gradually decreased after eCG injection, whereas NTSR1 expression remained unchanged, and NTSR2 expression was undetectable [42].

### 2.4. Spermatozoa and Testis

Hiradate et al. showed using Western Blot and immunohistochemistry the localization of NTSR1 in mice spermatozoa [29]. Using immunohistochemistry, they demonstrated that NTSR1 is more specifically localized in the neck of bull spermatozoa [28]. NTSR2 is present on their tail [30]. NTSR3 expression has not been studied in the sperm of cattle and mice.

In humans, NTSR1 was detected in spermatozoa using Western Blot techniques [46]. In non-human primates, immunohistochemistry detected NTSR1 at several stages of sperm maturation, with the strongest staining in mature spermatozoa in the lumen of the seminiferous tubules [46]. Spermatozoa and their precursors did not express NTSR2, but it was found in the interstitial cells of the seminiferous tubes, unlike NTSR1. NTSR3 was detected using immunohistochemistry in the testis but not in male germ cells at any stage of maturation [46]. NTS secretion was not studied. 

All these data about the localization of NTS and NTS receptors are summarized in Figure 1 and Table 1.

## 3. Role in Ovulation

Several studies showed an increase in NTS after ovulation triggering. Wissing et al. showed in 2014 using a microarray analysis and RT-PCR that NTS mRNA expression increased 15,7-fold 36 h after hCG triggering in the granulosa cells (GCs) of women undergoing a long agonist IVF protocol [47]. In order to evaluate the role of NTS in ovulation, Al Alem et al. studied in woman and in female rats the expression levels of NTS in GCs after ovulation triggering via an hCG injection [27]. In woman, the dominant follicle was surgically excised prior to the LH surge (the preovulatory phase), or women were given 250 μg hCG to provoke ovulation triggering, and dominant follicles were collected 12–18 h after hCG (the early ovulatory phase), 18–34 h (the late ovulatory phase), and 44–70 h (the postovulatory phase). NTS mRNA was massively increased during the early and late ovulatory phases in GCs (15,000-fold) and theca cells (700-fold) [27]. Moreover, in cultured granulosa–lutein cells (GLCs) from IVF patients (after 6 or 7 days with the aim of regaining their responsiveness to hCG), NTS expression was induced 6h after hCG treatment [27]. In cultured rat GCs treated with or without 1 IU hCG for 4, 8, 12, or 24 h, the expression of NTS increased after 4 h of hCG treatment by approximately 7-fold and remained elevated after 8 h, after which NTS expression declined to basal levels by 24 h [27]. Likewise, Campbell et al. demonstrated in female macaques that all the granulosa cells of the ovulatory follicle secrete NTS in an increased manner after the LH surge (up to 30 times more). This secretion was not found in theca cells [26]. 

The NTS regulation of ovulatory mechanisms has been studied in vitro. In both women and rats, the addition of an EGF pathway inhibitor (AG1478) partially blocked the expression of NTS mRNA by granulosa cells after an injection of hCG [27]. In humans, the regulation of NTS expression was blocked by the addition of inhibitors of the PKA and PKC pathways, as well as IP3 kinase and MAPK, suggesting that its regulation occurs via these pathways [27]. In mice, Shrestha et al. confirmed the mediation of NTS expression by PKA, MAPK, and EGF receptor signaling pathways and found several novel genes regulated by NTS (Ell2, Rsad2, Vps37a, and Smtnl2) that could impact the ovulatory process [42].

In order to demonstrate the role of the NTS/NTSR3 complex in ovulation, Campbell et al. inhibited NTS in vivo [26]. Over the monitored cycles, an intrafollicular injection of a rabbit antibody against NTS (ImmunoStar, Hudson) or control IgG was performed during aseptic surgery. Immediately postoperatively, hCG was administered to initiate ovulatory events. They showed that three out of four “anti-NTS-injected” follicles did not break without oocyte release and had an aspect of hemorrhagic cyst, whereas all the control ones showed a clear rupture of the membrane (four/four) [26]. The follicles injected with either the control IgG or the NTS antibody showed evidence of structural luteinization. In addition, both treatments resulted in similar levels of serum progesterone after hCG administration [26]. Finally, NTS stimulated ovarian microvascular endothelial cell migration in a dose-dependent manner but did not alter proliferation [26]. The mechanisms leading to follicular rupture through the interaction of NTS with NTSR3 are not known. However, it has been described in other models, notably in cancer, that the interaction of NTS with NTSR3 activates signaling cascades through focal adhesion kinase (FAK), a key pathway leading to the weakening of cell–cell and cell–extracellular matrix adhesions, a series of events that could be responsible for migration and cancer metastasis. Finally, some future approaches targeting NTSR3 release through the inhibition of matrix metalloproteinases (MMPs) are suggested [48].

These mechanisms leading to cell disjunction could explain the rupture of the follicular membrane during ovulation, showing a major role of NTS/NTSR3 in the mechanisms of ovulation (Figure 2).

## 4. Role in Sperm and Fertilization

### 4.1. Sperm Viability

Sperm viability was studied on human and non-human primates’ spermatozoa using eosin–nigrosin staining [46]. Treatment with 0.1, 1, or 10 μM NTS did not change sperm viability after 5 or 15 min of treatment compared to a control. Treatment for 30 min did result in a small but significant increase in sperm viability when compared to the control [46].

### 4.2. Sperm Motility

Campbell et al. analyzed the effects of adding NTS to a medium on sperm motility using a computer-assisted sperm analysis (CASA) system. Treatment with NTS did not modify the percentage of motile or progressive sperm or the percentage of sperm scored as rapid, slow, or static, but treatment with NTS at 1 and 10 μM did slightly but significantly increase the percentage of sperm scored as medium speed [46]. The same experiment has been used to study sperm motility in Japanese black cattle. No great effect on sperm motility has been shown, but a small number of sperm cells showed a hyperactivated motility pattern, suggesting that the effect of NTS on sperm motility is small, while cellular responses can be heterogeneous [28].

### 4.3. Capacitation

Capacitation is required to render spermatozoa competent to fertilize oocytes by passing through the female genital tract. It has been shown by Visconti et al. that the tyrosine phosphorylation of many proteins is induced during sperm capacitation [49]. Using a Western Blot analysis, Hiradate et al. showed that NTS can facilitate sperm capacitation in mice by enhancing this tyrosine phosphorylation in a dose-dependent manner in vitro [29]. Using the same technique, these results were confirmed in cattle, where total tyrosine phosphorylation was significantly increased upon the addition of NTS to the medium [28].

### 4.4. Acrosome Reaction

In human and non-human primates, Campbell et al. studied the NTS effects on the spermatozoa acrosomal reaction using immunofluorescence [46]. They demonstrated that NTS increased the acrosome reaction in a dose-dependent manner. Sperm incubated for 5 min with 0.1–10 μM NTS showed a dose-dependent decrease in complete acrosomes (*p* < 0.05), as well as a dose-dependent increase in absent acrosomes (*p* < 0.05), with no change in partial acrosomes, whereas sperm incubated for 15 min or 30 min showed a decrease in complete acrosomes (*p* < 0.05) and an increase in both partial and absent acrosomes (*p* < 0.05) [46]. They also proved that the action of NTS on the acrosome reaction in human spermatozoa was through its interaction with NTSR1. Indeed, the acrosome reaction appeared when NTS was added in the medium and was absent when they used SR48692, a specific antagonist of the NTS-NTSR1 complex [46]. Hiradate et al. had the same results using capacitated sperm from mice, which were incubated with increased concentrations of NTS to study its impact on the acrosome reaction. By using (Alexa Fluor) fluorescein staining, they showed that NTS treatment significantly accelerated the acrosome reaction in a dose-dependent manner [29]. When adding NTSR1 or NTSR2 antagonists (SR48692 and levocabastine, respectively) to the medium, the acrosomal reaction was completely blocked, suggesting the functional expression of not only NTSR1 but also of NTSR2 on sperm cells and the contribution of these receptors to the acrosome reaction [29]. Moreover, when 10 or 50 μM NTS was added, [Ca^2+^]_i_ levels were increased just after the addition. The calcium intensity when 50 μM of NTS was added was higher than that when 10 μM was added, providing evidence for dose dependency in the [Ca^2+^]_i_ response. However, the number of sperm cells that immediately increased their calcium levels were low at 50 μM (10 spermatozoa per 29 total examined) and 10 μM (3 spermatozoa per 50 total examined), whereas almost all the sperm experienced increased calcium mobilization when ionomycin was added (24 spermatozoa per 25 total examined) [29]. The same mechanisms appear to exist in bull semen. However, this reaction only concerns a few spermatozoa (only 10 of 29 observed), and this was observed with the highest dose used (50 µM of NTS) [28].

The potential role of NTS in spermatozoa is illustrated in Figure 3.

### 4.5. Fertilization

Fertilization was specifically studied in only one publication. The NTS pretreatment of monkey sperm reduced the fertilization rate of monkey MII oocytes in vitro from 72% [46]. Moreover, oocytes fertilized with untreated sperm were generally healthy in appearance, with the development of two pronuclei, whereas oocytes fertilized with NTS-treated sperm did not typically form pronuclei [46].

### 4.6. Embryo Development

Two publications studied the role of NTS in embryo development. The addition of NTS to an IVF medium significantly improved the levels of cleavage in the bovine embryo (36.5% to 50% of early cleavage and 49% to 64.4% of cleavage, *p* < 0.05) [30]. However, NTS had no impact on the number of blastocysts on days 7 and 8 of their experience. A greater number of cells in bovine blastocysts appears to be an advantage in improving the length of gestation and in fetal development in cattle [50]. To assess the quality of embryos, Umezu and his team counted embryo cells using an immunofluorescence technique to distinguish trophectodermal cells and inner mass cells. The blastocysts obtained with the addition of NTS to an IVF medium contained more cells in total, as well as in their inner mass cells, than the ones obtained from the IVF medium without NTS [30]. The effects of NTS were also studied in mice embryos before implantation. Hiradate et al. showed using RT-PCR that mice embryos expressed NTSR 1 and NTSR3 at the zygote stage. Their expressions decreased following the two-cell stage but remained detectable until the blastocyst stage [31]. Embryos were cultured in the presence of different concentrations of NTS for 96 h. There were no significant differences in the two-cell cleavage and four-cell rate between control and NTS treatment (1, 10, 100, and 1000 nM) groups, but a significant increase in blastocyst formation was observed in the 100 nM NTS group (60.6 ± 1.7% vs. 75.6 ± 3.4%, *p* = 0.03). In this treatment group, the hatching rate was 28.1 ± 3.8% versus that of the control group (13.0 ± 4.1%; *p* = 0.08) [31]. No data concerning oocyte quality related to embryo development were available.

## 5. Role in Implantation

### 5.1. Effects on Uterine Vasculature

The role of NTS in reproduction was first discussed in 1981 by Clark et al., who analyzed the effect of NTS on blood flow in the uterus of non-pregnant sheep [40]. They compared its effect to that of prostacyclin, which is one of the most potent vasodilator prostaglandins. Their study consisted of injecting neuropeptides directly using a catheter into the main uterine artery. They showed no vasoactive activity for NTS.

### 5.2. Effects on Oviduct Contractility

In 1987, Reinecke et al. showed using an isometric force displacement transducer that NTS enhanced autonomous oviduct contractility up to 55% in vitro and that their amplitude was 9-fold more than that without NTS addition [39].

### 5.3. Effects on Endometrium Receptivity

The effects of NTS on endometrial receptivity appear to be very different depending on the species. In dairy goats, Zhang et al. analyzed the concentrations of NTS mRNA and other molecules in the receptive endometrium and the pre-receptive endometrium [38]. Using inhibitory RNA systems, it was demonstrated that NTS was expressed 195 times more in the receptive endometrium than in the pre-receptive endometrium. To go further in their experiments on the potential functions of NTS in the receptive endometrium, they synthesized siRNA-NTS and a pcDNA3.1 (+) NTS vector to explore the effects of NTS in endometrial epithelial cells (EECs). With this experiment, they showed that the overexpression of NTS increased the proliferation of EECs (*p* < 0.005), while siRNA-NTS induced the apoptosis of EECs (*p* < 0.005). In addition, NTS may play a role in endometrial receptivity by increasing the levels of certain biochemical markers of the receptive endometrium, such as leukemia inhibitory factor (LIF), COX2, and HOXA10, in EECs in vitro [38]. However, these results seem contradictory with the results of Sakumoto et al. in cows, where the expression of NTS in the endometrium was higher in the summer when fertility levels were lower [32]. Conflicting results also seem to be observed in rats, where an intra-uterine microinjection of NTS on day 4 or 5 of pregnancy decreases in a significant manner the number of fetuses and the weight and glycogen of the uterus, whereas performing the same experiment on days 8, 9, and 10 or on days 14, 15, and 16 has no consequences on fetuses’ viability [37]. 

The NTS physiological functions in mammalian reproduction are summarized in Table 2.

## 6. Conclusions and Perspectives

Data regarding the important role of NTS and its receptors are now well-established. About 15 publications that we reviewed exhaustively show the action of NTS in ovulation and fertilization.

The data concerning ovulation are still incomplete. The exponential presence of NTS in granulosa cells, especially after LH, hCG, or eCG triggering, and the co-expression of the specific receptor NTSR3 point us towards an autocrine and/or paracrine mode of action. The limited experimental data support a necessary but not sufficient role in causing ovulation. Mechanistic studies conducting the inhibition testing of signaling pathways confirm a direct action via the MAPK pathway. 

Likewise, the presence of NTS in the female genital tract, particularly after ovulation, indicates a physiological role. Data concerning the stages of fertilization, as well as the presence of NTSR1 and NTSR2 on the sperm membrane (while they are absent from the epithelial cells of the female tract outside the pathological situation), could suggest a major paracrine effect presiding over capacitation and the acrosomal reaction. However, fertilization rates with NTS-treated sperm were lower than those obtained with untreated sperm. It would be interesting to experiment to determine whether these fertilization rates remain low if NTS is added directly to the fertilization medium rather than to the sperm preparation, as the NTS-induced acrosomal reaction should be performed as close to the oocyte as possible. NTS could potentially play a role in IVF in cases of fertilization failure.

The role of NTS in endometrial receptivity is not clear across species, yet implantation failures are a major cause of IVF failure, and its potential mechanisms are still poorly understood. The study of NTS expression at the endometrial level in women could be of interest in patients presenting repeated implantation failures after embryo transfer in this context.

## Figures and Tables

**Figure 1 ijms-24-04594-f001:**
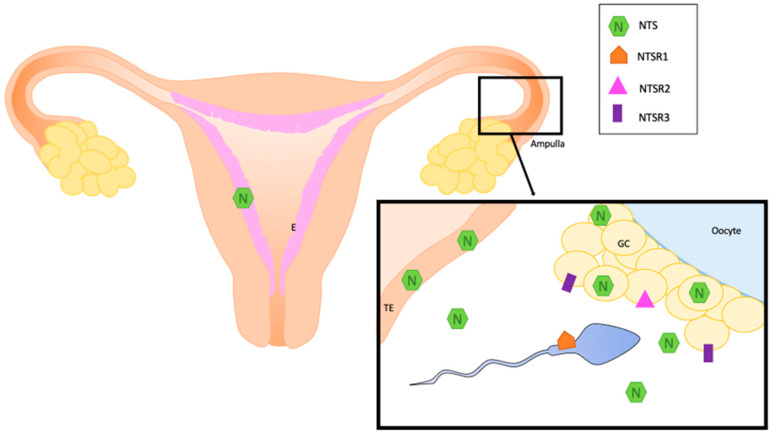
Localization of NTS and its receptors in normal human reproductive system. E: endometrium, GC: granulosa cells, NTS: neurotensin, NTSR1; neurotensin receptor 1, NTSR2: neurotensin receptor 2, NTSR3: neurotensin receptor 3, TE: oviduct epithelium.

**Figure 2 ijms-24-04594-f002:**
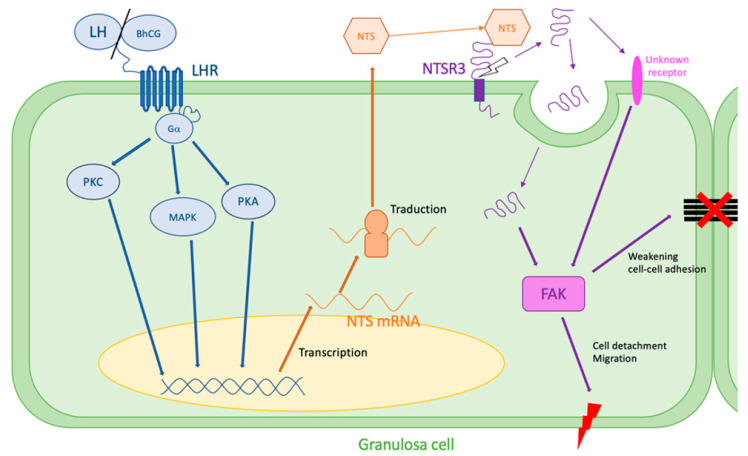
Hypothesis of NTS action during ovulation. LH surge stimulates NTS expression by the granulosa cells via MAPK, PKA, and PKC pathways. NTS acts on NTSR3 in an autocrine way. NTSR3 stimulates FAK pathway, which weakens cell–cell and cell–extracellular matrix adhesions and can lead to follicular rupture. FAK: focal adhesion kinase, LH: luteinizing hormone, LHR: LH receptor, MAPK: mitogen-activated protein kinase, PKA: phosphokinase A, PKC: phosphokinase C, NTS: neurotensin, NTSR3: sortilin.

**Figure 3 ijms-24-04594-f003:**
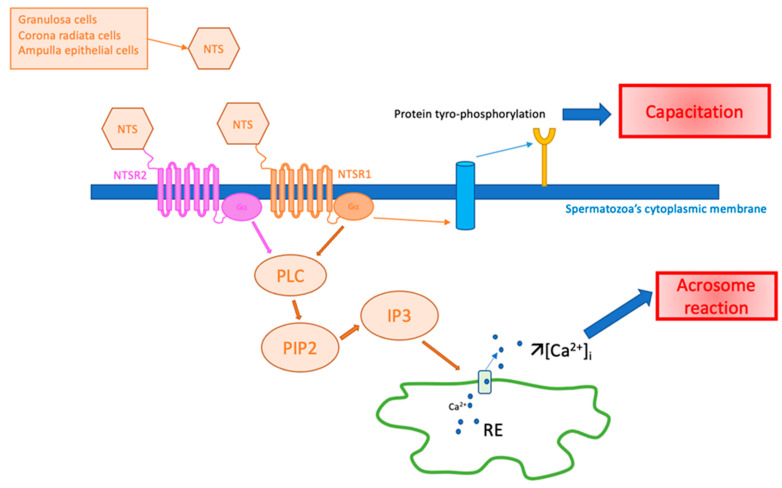
Hypothesis of NTS action on spermatozoa in fertilization. Ca^2+^: calcium ion, [Ca^2+^]_i_: intracytoplasmic calcium concentration, IP3: inositol triphosphate, NTS: neurotensin, NTSR1: neurotensin receptor 1, NTSR2: neurotensin receptor 2, PIP2: phosphatidyl-inositol 4,5, PLC: phospholipase C, RE: endoplasmic reticulum.

**Table 1 ijms-24-04594-t001:** Physiological expressions of NTS and its receptors in reproductive system in mammalians.

Tissue	Model	NTS	NTSR1	NTSR2	NTSR3	References
**Oviduct epithelium**	Cows	+	NA	NA	NA	[28,43]
	Mice	+	NA	NA	NA	[29,31]
	Goats	+	NA	NA	NA	[38]
	Women	+	NA	NA	NA	[39]
**Endometrium**	Cows	+	NA	NA	NA	[28,32]
	Mice	+	NA	NA	NA	[29,31]
	Goats	+	NA	NA	NA	[38]
	Women	+	-	NA	NA	[24]
**Granulosa cells**	Cows	NA	-	-	NA	[30]
	Mice Before OT After OT	+ +++	+ +	+/- -	+++ +	[29,42]
	Rats Before OT After OT	+ +++	+ +	+ +	+++ +++	[27]
	Macaques Before OT After OT	+/- +++	- -	+++ +/-	+/- +++	[26]
	Women Before OT After OT	+ +++	+/- +/-	+ +/-	+++ +	[27,47]
**Testis**	Macaques	-	+	+	-	[46]
	Bulls	-	+	+	-	[30]
**Spermatozoa**	Mice	-	+ (neck)	+	-	[29]
	Bulls	-	+ (neck)	+ (tail)	-	[28,30]
	Monkeys Men	- -	+ +	- -	- -	[46] [46]

OT: ovulation trigger, NTS: neurotensin, NTSR1: neurotensin receptor 1, NTSR2: neurotensin receptor 2, NTSR3: neurotensin receptor 3, -: not expressed, +/-: low expression to undetectable, +: expression, +++: strong expression, NA: not available.

**Table 2 ijms-24-04594-t002:** NTS physiological functions in mammalian reproduction.

Function	Models	Data	Ref
**Ovulation**	Macaques Rats Humans Mice	NTS is involved in follicular rupture in vivo NTS stimulates endothelial cell migration and capillary sprouts in vitro No effect on luteinization Through its increasing expression following OT, by PKA and PKC pathway along with PI3 kinase and MAPK	[26] [27] [27] [42]
**Sperm viability**	Macaques Humans	NTS slightly increases sperm viability (incubation in vitro >30 min)	[46] [46]
**Sperm motility**	Bulls Humans	No effect in vitro	[28] [46]
**Capacitation**	Mice Bulls	NTS facilitates sperm capacitation by enhancing tyrosine phosphorylation in vitro	[29] [28]
**Acrosome reaction**	Mice Bulls Macaques Humans	NTS increases intracellular calcium concentration (mice) NTS enhances acrosome reaction in a dose-dependent manner by interacting with NTSR1 and/or NTSR2 (mice)	[29] [28] [46] [46]
**Fertilization**	Macaques	NTS sperm pretreatment decreases fertilization rate	[46]
**Embryo development**	Cattle Mice	NTS improves embryonic cleavage rates in vitro No impact on blastocyst development rate More blastocyst cells in embryos NTS increases blastocyst development rate in vitro No effect on blastocyst cell number	[30] [29] [31]
**Endometrial receptivity**	Goats Rats	High NTS expression is associated with a receptive endometrium NTS increases endometrial epithelial cell proliferation and inhibits their apoptosis NTS increases the expressions of certain markers of endometrial receptivity, such as LIF, COX2, HOXA10 Intrauterine microinjection of NTS decreases the number of viable fetuses on day 4 or 5 of pregnancy and had no impact on days 8-16	[38] [37]

COX2: cyclooxygenase 2, HOXA10: homeobox protein A10, LIF: leukemia inhibitor factor, MAPK: mitogen-activated protein kinase, NTS: neurotensin, NTSR1: neurotensin receptor 1, NTSR2: neurotensin receptor 2, OT: ovulation trigger, PKA: protein kinase A, PKC: protein kinase C.

## Data Availability

Not applicable.

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
