# Peer review of "Neurotensin and Its Involvement in Reproductive Functions: An Exhaustive Review of the Literature"

_ijms, 2023, doi:10.3390/ijms24054594_

Round 1

Reviewer 1 Report

The present review of literature talks about “The implications of Neurotensin on reproductive functions). This research provides interesting information. However, it is necessary to make some changes before its final publication.

INTRODUCTION

General comments: I recommend adding more to this section explaining "Neurotensin" and its general implications on reproduction.

Specific comments:

Line 43-48.- I recommend changing this paragraph “NTS has a physiological role and is involved in several functions, mainly in digestive and central nervous systems. In digestive system, NTS acts on intestinal motility [10] and facilitates fat acid translocation [11]. In central nervous system, NTS modulates dopaminergic, serotonergic, GABAergic, glutamatergic, and cholinergic systems [12]. It plays a main role in decreasing body temperature and sleeping [13]. NTS also presents antinociceptive effects mediated by NTSR2 [14,15]” to line 32 because they are more general comments of “Neurotensin”

Line 65.- Better describe "the purpose of this review". I mean, describe all the objectives of this review

23. Cumulus and granulosa cells

General comments: in this section mention is made of the effect of hCG on granulosa cells but there is no evidence that another gonadotropin such as eCG has any effect? The following article was published “Shrestha, K., Al-Alem, L., Garcia, P., Wynn, M. A., Hannon, P. R., Jo, M., & Curry Jr, T. E. (2023). Neurotensin expression, regulation, and function during the ovulatory period in the mouse ovary. Biology of Reproduction, 108(1), 107-120.” He mentions the effect of eCG before PMSG on Neurotensin expression. I suggest reading it.

Line 106.- I suggest describing all the “MAPK” abbreviations. Are you referring to “mitogen-activated protein kinases”? define it.

Line 150.- I suggest defining "EGF" because it is defined up to line 167 "epidermal growth factor (EGF)"

3.Role in ovulation

Line 155.- I suggest eliminating "(GC)" as described in line 152

Line199.- Figure 2. I suggest increasing the font size in the figure, it is a bit difficult to distinguish what it says.

4. Role in sperm and fertilization

Suguero increase sections “4.1. Sperm viability and 4.2. Sperm motility”

Line 211. What type of sperm motility? Progressive, individual, etc.

4.6. embryo development

General comments: Check the format, there is a lot of space between lines that does not correspond to the other paragraphs

Author Response

We thank the reviewers for their work and comments which have significantly improved the content of our manuscript. We have answered to their request as following.

Response to reviewers:

Reviewer 1

“INTRODUCTION General comments: I recommend adding more to this section explaining "Neurotensin" and its general implications on reproduction.

Thank very much for your comments. We interestingly supplemented the introduction with general implications on reproduction (page 2; lines 65-67).

Specific comments:

“Line 43-48.- I recommend changing this paragraph “NTS has a physiological role and is involved in several functions, mainly in digestive and central nervous systems. In digestive system, NTS acts on intestinal motility [10] and facilitates fat acid translocation [11]. In central nervous system, NTS modulates dopaminergic, serotonergic, GABAergic, glutamatergic, and cholinergic systems [12]. It plays a main role in decreasing body temperature and sleeping [13]. NTS also presents antinociceptive effects mediated by NTSR2 [14,15]” to line 32 because they are more general comments of “Neurotensin””

We changed this paragraph to line 32 and updated the references.

“Line 65.- Better describe "the purpose of this review". I mean, describe all the objectives of this review.”

We have specified the objectives of the review (page 2; lines 75-80).

“2. Cumulus and granulosa cells.

General comments: in this section mention is made of the effect of hCG on granulosa cells but there is no evidence that another gonadotropin such as eCG has any effect? The following article was published “Shrestha, K., Al-Alem, L., Garcia, P., Wynn, M. A., Hannon, P. R., Jo, M., & Curry Jr, T. E. (2023). Neurotensin expression, regulation, and function during the ovulatory period in the mouse ovary. Biology of Reproduction, 108(1), 107-120.” He mentions the effect of eCG before PMSG on Neurotensin expression. I suggest reading it.”

We had already integrated this article (reference 42). We have indeed corrected hCG by eCG (page 2; lines 89-90 and page 3; line 138).

“Line 106.- I suggest describing all the “MAPK” abbreviations. Are you referring to “mitogen-activated protein kinases”? define it.”

We defined “MAPK” as “mitogen-activated protein kinases” (page 2; lines 47-48).

“Line 150.- I suggest defining "EGF" because it is defined up to line 167 "epidermal growth factor (EGF)"”

We defined “EGF” as “epidermal growth factor” (page 3; line 120).

“3. Role in ovulation.

Line 155.- I suggest eliminating "(GC)" as described in line 152.”

We eliminated this redundant mention.

“Line 199.- Figure 2. I suggest increasing the font size in the figure, it is a bit difficult to distinguish what it says.”

We’ve increased the size of the 3 figures.

“4. Role in sperm and fertilization

Suguero increase sections “4.1. Sperm viability and 4.2. Sperm motility”

Line 211. What type of sperm motility? Progressive, individual, etc.”

We specified the data concerning sperm motility (page 7; lines 228-234).

“4.6. embryo development

General comments: Check the format, there is a lot of space between lines that does not correspond to the other paragraphs.”

We erased the lots of space between lines; to maintain uniformity (pages 8-9; lines 274-292).

Reviewer 2 Report

The authors reviewed the neurotensin roles mainly in reproductive system, ovulation, in sperm and fertilization, and implantation. It would be great if the authors could find and include how neurotensin affects the oocyte quality and its subsequent fertilization and embryo development.

Author Response

We thank the reviewers for their work and comments which have significantly improved the content of our manuscript. We have answered to their request as following.

Response to reviewers:

Reviewer 2

“The authors reviewed the neurotensin roles mainly in reproductive system, ovulation, in sperm and fertilization, and implantation. It would be great if the authors could find and include how neurotensin affects the oocyte quality and its subsequent fertilization and embryo development.”

Data concerning oocyte quality are unfortunately not available in the articles. We specified it in the part "embryonic development" (page 9; lines 304).